# THE OUTER PRODUCT STRUCTURE OF NEURAL NETWORK DERIVATIVES

**Craig Bakker, Michael J. Henry & Nathan O. Hodas**
Pacific Northwest National Laboratory
Richland, WA 99352, USA
`{craig.bakker,michael.j.henry,nathan.hodas}@pnnl.gov`

## ABSTRACT

Training methods for neural networks are primarily variants on stochastic gradient descent. Techniques that use (approximate) second-order information are rarely used because of the computational cost and noise associated with those approaches in deep learning contexts. We can show that feedforward and recurrent neural networks exhibit an outer product derivative structure but that convolutional neural networks do not. This structure makes it possible to use higher-order information without needing approximations or significantly increasing computational cost.

## 1 INTRODUCTION

Gradient-based optimization methods use derivative information to choose search directions when minimizing a continuous objective function. The steepest descent method is the most basic of these techniques, but it converges very slowly in ill-conditioned systems. Newton's method uses second-order derivative information to take more efficient update steps, but it does not scale well and may fail if the Hessian is indefinite or singular. A variety of methods have been developed to try and appropriate the strengths of each approach while avoiding their weaknesses. In general, second-order information is hard to get but can significantly improve optimization performance. For further information about gradient-based optimization, see Nocedal & Wright (2006).

Deep learning (DL) provides problems that can be tackled with gradient-based optimization methods, but it has several unique features and challenges. DL problems can be extremely large and highly nonconvex, and training deep networks via mini-batch sampling results in a stochastic optimization problem; the variance associated with the batch sample calculations produces statistical noise, which can make it more difficult to perform the optimization. Deep networks also consist of the composition of known analytic functions. As such, we can calculate derivative information analytically via back-propagation. These special characteristics of DL have motivated researchers to develop training methods specifically designed to overcome the challenges with training neural networks (Kingma & Ba, 2014; Sutskever et al., 2013). Some researchers have used (approximate) second-order information to improve the training process, but sample noise can overwhelm the estimation or require special modifications to the method to prevent divergence (Byrd et al., 2016; Mokhtari & Ribeiro, 2014; Moritz et al., 2016; Schraudolph et al., 2007).

## 2 THE OUTER PRODUCT STRUCTURE OF NEURAL NETWORK DERIVATIVES

Second-order derivatives are not widely used in DL, and where they *are* used, they are typically estimated. These derivatives can be calculated analytically, but this is rarely done because of the scalability constraints described in Section 1. However, we can show that neural network derivatives have an outer product structure. When a matrix has this structure, it means that the information contained in that matrix (or the operations performed by that matrix) can be fully represented without needing to know every entry of that matrix; the same applies for higher-order tensors with outer product structure. We can therefore calculate, store, and use higher-order derivatives exactly in an efficient manner by only dealing with the outer product components.

### 2.1 Feedforward and Recursive Neural Networks

As an example, we illustrate the outer product structure for a feedforward network, of arbitrary depth and layer widths, consisting of ReLUs in the hidden layers, an arbitrary output layer activation, and an arbitrary error function. A feedforward network with arbitrary activation functions has more complicated derivative formulae, but those derivatives still exhibit an outer product structure. Table 1 provides a nomenclature for our deep network definition.

Table 1: Nomenclature – Formulation

| Quantity | Description |
|---|---|
| $n$ | Number of hidden layers |
| $v^{(0),i} = x^i$ | Vector of inputs for a single sample |
| $v^{(k),j} = \mathcal{A}\left(w_i^{(k),j} v^{(k-1),i}\right)$ | Vector output of layer $k$ |
| $w_i^{(k),j}$ | Matrix of weights for layer $k$ |
| $\mathcal{A}\left(\cdot\right)$ | Activation function |
| $u_i^j$ | Matrix of output layer weights |
| $p^j = u_i^j v^{(n),i}$ | Vector of intermediate variables for the output layer |
| $y^l$ | Vector of labels for a single sample |
| $f = f\left(y^l, p^j\right)$ | Scalar objective function value for a single sample |

Our calculations make extensive use of index notation with the summation convention (Ivancevic & Ivancevic, 2007) because index notation makes outer product structures clear – especially for higher-rank tensors. For example, if $A_{ij} = a_i b_j$, then $A_{ij}$ has an outer product structure. In index notation, a scalar has no indices ($v$), a vector has one index (**v** as $v^i$ or $v_i$), a matrix has two (**V** as $V^{ij}$, $V_j^i$, or $V_{ij}$), and so on. The summation convention holds that repeated indices in a given expression are summed over unless otherwise indicated. For example, $\sum_i a^i b_i = a^i b_i$. Here, we have adapted index notation slightly: indices in brackets (e.g. the $k$ in $v^{(k),j}$) are only summed over when explicitly indicated by a summation sign. We also use the Kronecker delta ($\delta^{ij}$, $\delta_j^i$, or $\delta_{ij}$), which is the identity matrix represented in index notation; it is 1 for $i = j$ and 0 otherwise. The first derivatives, on a per-sample basis, for this deep network are

$$\gamma_s^{(k),j} \equiv \delta_s^j \mathcal{A}'\left(w_i^{(k),j} v^{(k-1),i}\right) \tag{1}$$

$$\beta_i^{(k),j} \equiv \mathcal{A}'\left(w_l^{(k),j} v^{(k-1),l}\right) w_i^{(k),j} = \gamma_s^{(k),j} w_i^{(k),s} \tag{2}$$

$$\alpha_{j_l}^{(k,l),j_k} \equiv \begin{cases} \prod\limits_{i=l+1}^{k} \beta_{j_{i-1}}^{(i),j_i} & k > l \\ \delta_j^i & k = l \\ 0 & k < l \end{cases} \tag{3}$$

$$\eta_i^{(k,l),j} \equiv \alpha_s^{(k,l),j} \gamma_i^{(k),s} \tag{4}$$

$$\frac{\partial f}{\partial u_j^i} = \frac{\partial f}{\partial p^i} v^{(n),j} \tag{5}$$

$$\frac{\partial f}{\partial w_j^{(k),i}} = \frac{\partial f}{\partial p^l} u_m^l \eta_i^{(n,k),m} v^{(k-1),j} = \left(\frac{\partial f}{\partial p^l} u_m^l \eta_i^{(n,k),m}\right) \times v^{(k-1),j} \tag{6}$$

In calculating these expressions, we have deliberately left $\frac{\partial f}{\partial p^j}$ unevaluated. This keeps the expression relatively simple, and programs like TensorFlow (Abadi et al., 2015) can easily calculate this for us. Leaving it in this form also preserves the generality of the expression – there is no low-rank structure contained in $\frac{\partial f}{\partial p^j}$, and the low-rank structure of the network as a whole is therefore shown to be independent of the objective function and the output layer activation function. Index notation makes the outer product structure in Equations 5 and 6 clear; in Equation 6, the component in brackets is the product of several matrix multiplications that finally result in a vector (indexed by $i$). The second-order objective function derivatives are then

$$\frac{\partial^2 f}{\partial u_j^i \partial u_t^s} = \frac{\partial^2 f}{\partial p^i \partial p^s} v^{(n),j} v^{(n),t} \tag{7}$$

$$\frac{\partial^2 f}{\partial u_j^i \partial w_t^{(k),s}} = \frac{\partial f}{\partial p^i} \eta_s^{(n,k),j} v^{(k-1),t} + \frac{\partial^2 f}{\partial p^i \partial p^l} v^{(n),j} u_m^l \eta_s^{(n,k),m} v^{(k-1),t} \tag{8}$$

$$\frac{\partial^2 f}{\partial w_j^{(k),i} \partial w_t^{(r),s}} = \frac{\partial^2 f}{\partial p^l \partial p^q} u_m^l \eta_i^{(n,k),m} v^{(k-1),j} u_a^q \eta_s^{(n,r),a} v^{(r-1),t}$$

$$+ \frac{\partial f}{\partial p^l} u_m^l \times \begin{cases} \eta_s^{(n,r),m} \eta_i^{(r-1,k),t} v^{(k-1),j} & r > k \\ 0 & r = k \\ \eta_i^{(n,k),m} \eta_s^{(k-1,r),j} v^{(r-1),t} & r < k \end{cases} \tag{9}$$

Calculating all of these second derivatives requires the repeated use of $\frac{\partial^2 f}{\partial p^i \partial p^j}$. Evaluating that Hessian is straightforward given knowledge of the activation functions and objective used in the network, and storage is simple as long as the number of categories is small relative to the number of weights. The key thing to note about these second derivatives is that they retain a low-rank structure – they are now tensor products (or the sums of tensor products) of matrices and vectors.

This low-rank structure only exists *for each sample*. Even if the gradient of $f$ is an outer product, the gradient of the expectation, $E[f]$, will not be, because the gradient of $E[f]$ is the weighted sum of the gradients of $f$. However, if the mini-batch size is small relative to the number of weights, then the gradient of $E[f]$ will be low-rank and it may still be more efficient to work with the outer product components than with the gradient as a whole. The same goes, *mutatis mutandis*, for the Hessian: manipulating the entire Hessian will depend on how large the mini-batch size is relative to the number of weights and on the possible existence of a tensor inverse. Given this, the computational savings provided by using this structure will be most salient when calculating scalar or vector quantities on a sample-by-sample basis and then taking a weighted sum (such as an expectation) of the results.

Recurrent neural networks have derivative formulae that are very similar to those shown here, and those derivatives exhibit a similar outer product structure.

## 2.2 Convolutional Neural Networks

Convolutional layers do not destroy the outer product structure of subsequent feedforward or recurrent layers' derivatives, but they themselves do not have an outer product structure. In fact, a convolutional layer is a different way of using low-rank structure. A convolutional layer is equivalent to a feedforward layer where the layer weights form a doubly block circulant matrix (Goodfellow et al., 2016). The information contained in that (very large) matrix can be completely expressed by a much smaller matrix – the convolution kernel, in fact. This kernel enforces structure on weights and their derivatives, but the outer product structure on feedforward and recurrent layers only applies to the derivatives; convolutional layers tend to have fewer weights as a result. Both types of structure, however, make it possible to represent derivative information in a compact fashion.

## 3 Discussion and Conclusions

We can show that neural network derivatives have an outer product structure, and this structure makes it feasible to use higher-order derivative information when training the network. Machine learning research involving Sufficient Factor Broadcasting has referred to this outer product structure in certain contexts and with the goal of speeding up existing computational methods (Xie et al., 2015; Zhang et al., 2017). Those papers have only claimed that this structure exists, however – they have not proven it or listed the conditions under which it holds. We show that this structure exists for feedforward and recurrent neural networks. Moreover, we can show that this structure extends to higher-order derivatives. The outer product structure may be useful in speeding up gradient calculations, but it also opens the door for employing higher-order derivative information in a tractable fashion even for large neural networks.

ACKNOWLEDGMENTS

This research was performed at the Pacific Northwest National Laboratory, a multi-program national laboratory operated by Battelle for the U.S. Department of Energy.

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
