# OpenReview forum: "The Outer Product Structure of Neural Network Derivatives"
_ICLR.cc/2018/Workshop — Reject_

### Official Review · AnonReviewer2 · 2018-03-09
**useful gradient calculation mechanism generalises to second order but lacks algorithm and experiments**

**Rating:** 5
**Confidence:** 4

**Review:**

This paper shows the outer-product structure in gradient, which has been reported by previous works, specifically the "sufficient-factor" work in the context of first-order gradient.  The authors show that similar structure exists in the second order gradient.

However, it is unclear how this structure can be applied to algorithms that speedups the update. I would encourage the author to come up with such algorithm and along with appropriate experiments to show the applicability of the method.

One potential drawback of such approach is that the factored representation relies on small batch size. While second order method usually requires bigger batch-sizes.

---

### Official Review · AnonReviewer1 · 2018-03-13
**Interesting, though opaque paper**

**Rating:** 6
**Confidence:** 4

**Review:**

The paper is well-written and well-motivated, but the central claim, "[feed-forward and recurrent neural networks] exhibit an outer product derivative structure", is not spelled out in sufficient mathematical clarity -- nor is the contrasting claim, that convolutional networks do not possess this structure. No experiments or specific applications of the finding are described in detail. In my view this would nonetheless be an interesting workshop presentation.

---

### Decision · Program_Chairs · 2018-03-20
**ICLR 2018 Workshop Acceptance Decision**

**Decision:**

Reject

**Comment:**

Based on the reviews, this paper has not been accepted for presentation at the ICLR workshop. However, the conversation and updates can continue to appear here on OpenReview.